# Mutual reinforcement of land-based carbon dioxide removal and international emissions trading in deep decarbonization scenarios

Jennifer Morris [1] ✉, Angelo Gurgel [1], Bryan K. Mignone [2], Haroon Kheshgi [3] & Sergey Paltsev [1]

Carbon dioxide removal (CDR) technologies and international emissions trading are both widely represented in climate change mitigation scenarios, but the interplay among them has not been closely examined. By systematically varying key policy and technology assumptions in a global energy-economic model, we find that CDR and international emissions trading are mutually reinforcing in deep decarbonization scenarios. This occurs because CDR potential is not evenly distributed geographically, allowing trade to unlock this potential, and because trading in a net-zero emissions world requires negative emissions, allowing CDR to enable trade. Since carbon prices change in the opposite direction as the quantity of permits traded and CDR deployed, we find that the total amount spent on emissions trading and the revenue received by CDR producers do not vary strongly with constraints on emissions trading or CDR. However, spending is more efficient and GDP is higher when both CDR and trading are available.

Climate stabilization pathways limiting warming to 1.5 °C or 2 °C typically rely on negative emissions to offset remaining positive greenhouse gas (GHG) emissions in hard-to-abate sectors[1]. Land-based carbon dioxide removal (CDR) technologies such as bioenergy with carbon capture and storage (BECCS) and afforestation and reforestation (A/R) are the CDR options most commonly deployed in mitigation scenarios[1,2]. While many studies have explored the potential role of BECCS and A/R, little attention has been given to how their deployment might interact with, or depend on, the assumed climate policy regime. Because regional comparative advantages in different CDR options are not evenly distributed, total CDR deployment may depend on the extent to which international greenhouse gas (GHG) permit trade occurs. The level of trading, in turn, may depend on how Article 6 of the Paris Agreement[3] is implemented. Conversely, if CDR is limited, international trade of GHG permits may become increasingly constrained as national climate targets approach net-zero emissions since net positive emissions in some regions would only be enabled by net negative emissions in other regions. In this work, we investigate the interplay between land-intensive CDR options and international trade

in GHG permits to assess the potential deployment of such technologies and the size of international carbon markets.

Different CDR options are characterized by differences in technological and institutional readiness, cost, use of land, permanence of $CO_2$ removal, and social acceptance[4,5]. These differences are likely to vary by region and over time. Many studies have investigated the potential for deployment of CDR options under climate policy. Some have focused on a portfolio of CDR options[6–9], with estimated maximum total CDR potential ranging from ~10 to 35 gigatonnes of $CO_2$ ($GtCO_2$) per year. Others have explored specific CDR options such as BECCS[10], A/R[11], direct air capture (DAC)[12], biochar[13], enhanced weathering[14,15] and ocean iron fertilization[16]. BECCS and A/R are the two most widely deployed CDR options in mitigation scenarios[2].

A/R is well known, has been already implemented, can sequester carbon at lower costs than many CDR options, and could help to reverse biodiversity losses and improve the provision of ecosystem services[17,18]. However, it takes decades to accumulate fixed carbon in living biomass and soils, with saturation of carbon uptake occurring over time[19,20]. In addition, reforested areas must be set aside to retain

[1]Massachusetts Institute of Technology, Cambridge, MA, USA. [2]ExxonMobil Technology and Engineering Company, Annandale, NJ, USA. [3]University of Illinois at Urbana-Champaign, Urbana, IL, USA. ✉e-mail: holak@mit.edu

fixed carbon, removing the option of other potential productive uses. At the same time, that land is subject to disturbances, such as fires and disease, which can disrupt the sequestered carbon stocks[20,21], and as such, the effectiveness of such options may depend on robust monitoring and verification.

On the other hand, BECCS uses a given amount of land to remove $CO_2$ indefinitely (as bioenergy is harvested and the $CO_2$ is sequestered annually) and can provide a valuable energy co-product[10]. Given recent attention to DAC, it is worth noting that BECCS is typically found to be more cost-effective than DAC as a carbon removal option[12]. However, BECCS is not widely deployed currently, and it is more costly than A/R. Biomass production associated with potential large-scale adoption of BECCS also raises environmental and socio-economic concerns related to land use changes, biodiversity loss, water use, and commodity prices, among others[22–26]. Second-generation energy crops and wood from managed forestry[27], as well as use of wastes and residues[28] may reduce sustainability issues, although the true potential of sustainable biomass remains uncertain.

Institutional and policy constraints, such as strong decarbonization goals and net-zero emissions targets, are key drivers of demand for negative emissions[29]. Consequently, the deployment of alternative CDR options at the regional and global levels depends on the climate policy framework. Article 6 of the Paris Agreement[30] enables cooperation among parties in implementing their nationally determined contributions (NDCs), including international trade in GHG permits, which can reduce overall abatement costs and improve economic efficiency[31,32]. As the Article 6.4 Supervisory Body sets out to develop the requirements and processes needed to operationalize an international carbon crediting mechanism, growing discussion has focused on how to incorporate negative emissions into such a framework, and in particular how to address differences in the permanence of different CDR options to ensure the equivalence of permits[33]. Potential options include discounting or weighting offsets according to their reversal risk[34] or climate repair value[5], buffer pools[35], insurance for offsets[36], and the like-for-like principle of matching the durability of a removal activity with the permanence of an emissions activity (for example, only allowing CDR from A/R to offset emissions from land use changes)[37]. While these details need to be resolved, there is general consensus that Article 6 can play an important role in reducing costs and improving environmental benefits of meeting Paris targets[38–40].

Despite the abundance of research focused separately on international emissions trading and on CDR, there is limited discussion in the literature about the interactions between the two. One exception is Fajardy and Mac Dowell[41], who found emissions trading to be important to BECCS deployment. However, that paper did not consider A/R and only considered how international emissions trading could affect BECCS deployment, not how BECCS deployment could affect international emissions trading.

In this work, we contribute to the literature by undertaking a structured analysis designed to investigate the two-way nature of the relationship between land-based CDR deployment and international emissions trading. We focus specifically on BECCS and A/R, and leave consideration of other land-based (e.g., biochar) and non-land-based (e.g., DAC) CDR options for future study. In our analysis, we take into account connections between energy and agricultural markets and endogenous land use competition in different regions of the world. Our approach explicitly represents key aspects of the energy and land systems required to assess the role of land-based CDR in deep decarbonization scenarios. These aspects include the costs of BECCS and biomass technologies, land conversion costs, $CO_2$ emissions and sequestration from direct and indirect land use changes, land availability and competition, international trade of agricultural and food commodities, and decarbonization policies, including their effects on technology competition, goods prices, and aggregate income and consumption. More details about the modeling framework are provided in the Methods section. We examine scenarios with and without international trade in GHG permits, which could reflect alternative future developments under Article 6 of the Paris Agreement, and with and without CDR available/covered under the emissions policy, reflecting both technology and policy uncertainty. Although these assumptions are rather extreme, they allow us to investigate drivers of differences in mitigation outcomes. We show that CDR deployment and international trade in permits are interdependent and can reinforce each other, suggesting that understanding CDR outcomes may require deeper understanding of international carbon markets and vice versa.

## Results

We consider four main scenarios to investigate BECCS and A/R deployment and the interactions with international trade in permits, all considering the same 2 °C temperature stabilization target (Table 1). For these scenarios, CDR availability refers only to BECCS and A/R. The regional emissions caps associated with the global 2 °C target are described in the Methods section and shown in Supplementary Fig. 2. The resulting global net GHG emissions path is shown in Supplementary Fig. 3. Sensitivity cases with a more stringent stabilization target, a higher emissions cap for China, and alternative BECCS and A/R assumptions are also included in the Supplementary Information.

### Global CDR and emissions trading outcomes

Taken together, the scenarios show that the availability of CDR substantially increases the scale of international emissions trading, and the presence of an international emissions trading regime substantially increases the scale of CDR (Fig. 1, also Supplementary Fig. 12). When CDR is deployed, it is used to offset fossil $CO_2$ and non-$CO_2$ GHG emissions that otherwise would need to be abated (see Supplementary Fig. 3). In terms of CDR deployment, A/R deploys first, followed by BECCS. While A/R deployment peaks before the end of the century, BECCS continues to grow throughout the century. By the end of the century, about two-thirds of cumulative CDR is from BECCS (see Supplementary Fig. 4).

On the one hand, trade unlocks the full economic potential of CDR across regions. On the other hand, as regional emissions targets approach net zero, international trade in permits is only possible with the deployment of CDR producing negative emissions that allow some regions to retain net positive emissions. As a result, global cumulative CDR from 2020 to 2100 under GHG trade is much larger than without GHG trade, and trade is much larger when CDR is deployed. While the use of international offsets in climate policy discussions has historically focused on nature-based CDR[34,42,43], the interplay between CDR and permit trade that we observe is more dependent on engineered CDR in the form of BECCS given that BECCS provides most of the cumulative CDR. These results align with the finding by Fajardy and Mac Dowell[41] that inter-regional trading is required to deploy the lowest-cost BECCS options globally. We extend that finding by also including A/R and by demonstrating the two-way nature of the relationship between CDR deployment and international emissions trading.

## Table 1 | Matrix of scenarios

| | | International emissions trading | |
|---|---|---|---|
| | | NO | YES |
| CDR available | NO | NoCDR_NoTrade | NoCDR_Trade |
| | YES | CDR_NoTrade | CDR_Trade |

CDR availability refers only to BECCS and A/R. No other CDR options are included in these scenarios.

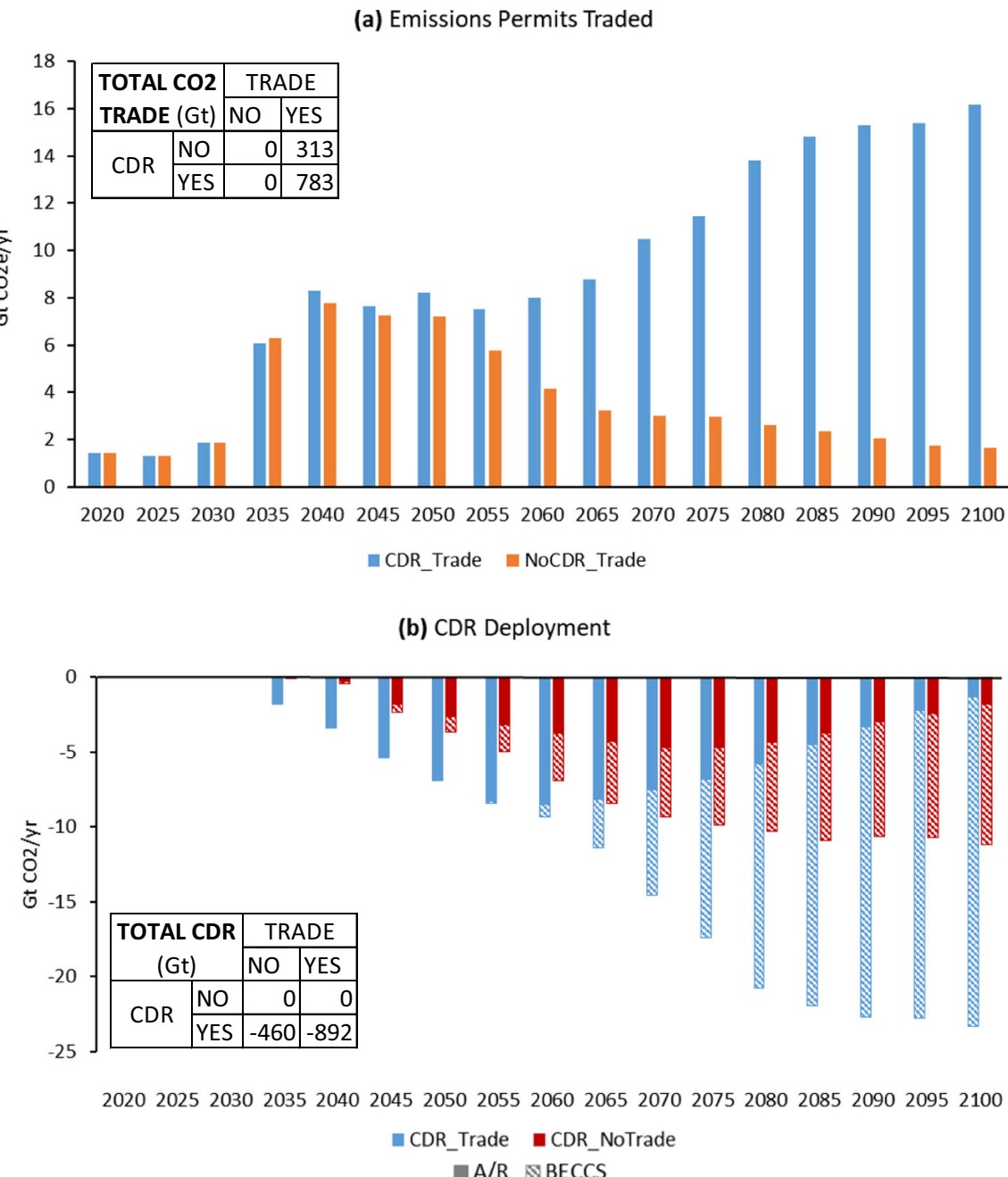

**Fig. 1 | The relationship between land-based carbon dioxide removal (CDR) and emissions trading in 2 °C scenarios. a** Greenhouse gas (GHG) emissions permits traded globally with vs. without Carbon Dioxide Removal (CDR); **b** CDR Deployment (Afforestation/Deforestation (A/R) and Bioenergy with carbon capture and storage (BECCS)) globally with vs. without international emissions permit trading. The inlaid tables in each panel are cumulative emissions from 2020 to 2100 that are traded (Total CO₂ Trade) and removed (Total CDR) in the different scenarios. A/R in panel (b) is defined as net negative land CO₂ emissions (net positive land use emissions in early years are not shown) and BECCS removals (land use change emissions excluded).

The deployment of BECCS and A/R is separated in time since each is deployed in a different carbon price range. A/R is deployed earlier due to lower costs. However, total A/R potential in each region is limited by the area ecologically suitable to grow forests[18]. In contrast, the cost of BECCS is higher, but its deployment is less constrained, and over time it can sequester more carbon per unit of land committed. In other words, in the long run, BECCS is more effective at removing carbon, since the annual amount of carbon removed for a given land area can persist indefinitely, while A/R is constrained to a maximum stock of carbon sequestered for the same land area[44]. However, we find that the roles of A/R and BECCS are robust, with only modest changes to the deployment of each under different assumptions about the availability or scalability of the other (Supplementary Figs. 9 and 11). Cumulatively, total $CO_2$ sequestration from CDR deployment between 2020 and 2100 is about 460 GtCO₂ in the CDR_NoTrade case and 900 GtCO₂ in the CDR_Trade case. As points of reference, for comparable 2 °C scenarios, the Intergovernmental Panel on Climate Change (IPCC) Special Report on Global Warming of 1.5 °C finds a range of 0 to over 1200 GtCO₂ cumulative CDR (with a median of ~500 GtCO2)[2]; the IPCC Sixth Assessment Report (AR6) finds 170–650 GtCO₂ from BECCS and 10–250 GtCO₂ from A/R[1]; and Fajardy et al.[10] finds ~620 GtCO₂ of BECCS.

These results assume equal treatment of CDR permits from A/R and BECCS. However, as mentioned, these two options have different

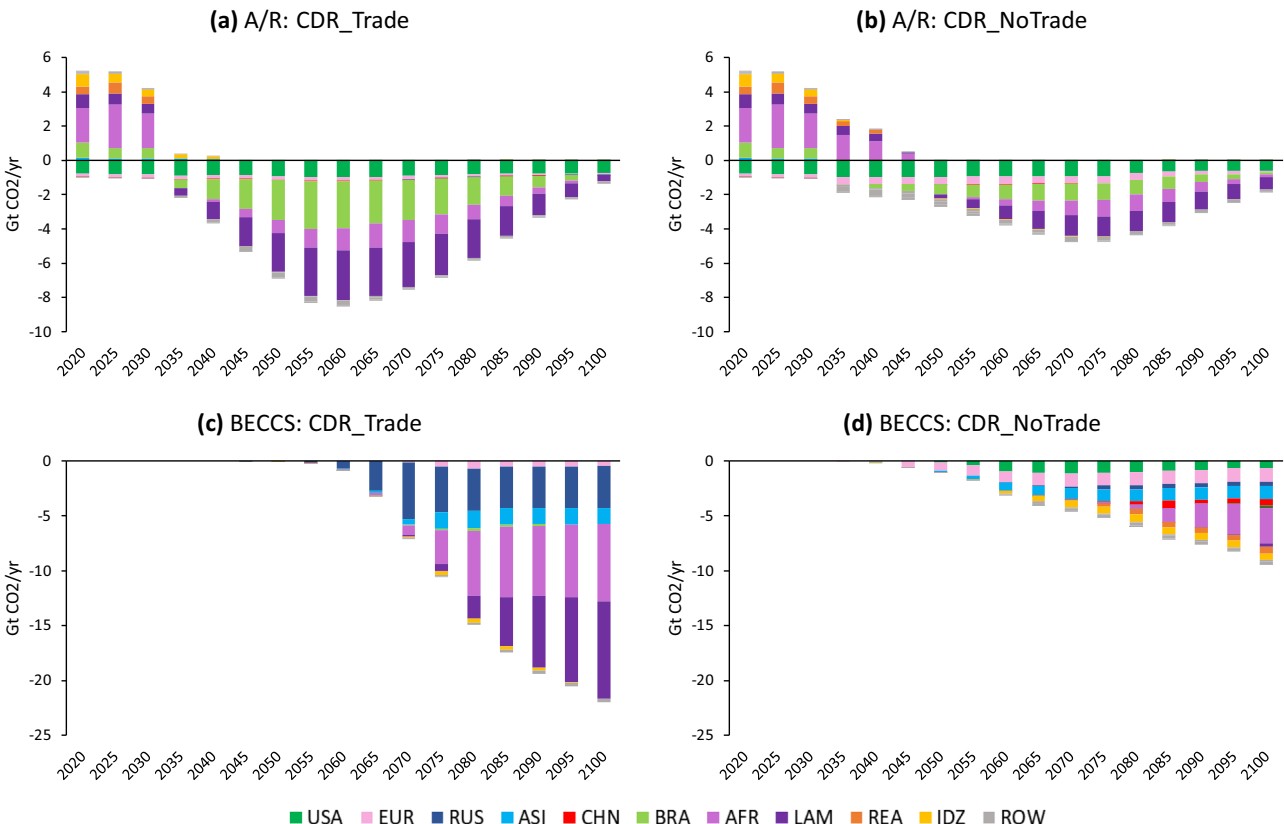

**Fig. 2 | Regional deployment of afforestation/reforestation (A/R) and bioenergy with carbon capture and storage (BECCS) under the different 2 °C scenarios. a, b** show net land use change emissions, which are positive in early years and negative in later years. **c, d** show removals from BECCS (land use change emissions excluded), with removals shown as negative values. USA United States,

EUR Europe (EU+), RUS Russia, ASI Dynamic Asia (Malaysia, Philippines, Singapore, Taiwan), CHN China, BRA Brazil, AFR Africa, LAM Other Latin America, REA Other East Asia, IDZ Indonesia, ROW Rest of World. Regional definitions are provided in Supplementary Fig. 1.

levels of sequestration permanence and reversal risks. In our sensitivity analysis (see Supplementary Table 1), we also include BECCSonly scenarios in which credits from A/R are not allowed. Scenarios with no credits for A/R and the CDR scenarios with equal credits for A/R and BECCS provide bounds on the treatment of A/R crediting, with more nuanced crediting rules related to the permanence of A/R falling in between. The treatment of A/R crediting affects the level of fossil $CO_2$ emissions (Supplementary Figs. 7 and 8), the trajectories of BECCS and A/R deployment (Supplementary Fig. 9), the amount of emissions permits traded (Supplementary Fig. 10a) and the CDR deployment (Supplementary Fig. 10b), especially around mid-century. However, the overall finding that land-based CDR deployment and international emissions trading are mutually reinforcing remains robust to the assumptions about A/R crediting (Supplementary Fig. 10).

### Regional differences
International emissions trading encourages more CDR in countries with comparative advantage in CDR production (especially related to land availability), and as such, the majority of CDR comes from a few regions: Latin America, Africa, Brazil, and Russia (Fig. 2). In these regions, the choice between BECCS or A/R (or both) is mostly dependent on the potential cumulative $CO_2$ sequestration and relative costs to implement each option. While Africa and Latin America largely implement both types of CDR, relatively lower A/R costs in Brazil favor A/R, whereas higher A/R costs in Russia favor the adoption of BECCS, particularly if trade is allowed in the case of Russia.

International trading enables the full economic potential of CDR in these regions, increasing the volume of traded GHG permits and overall CDR deployment, thereby providing less expensive mitigation

to regions with more stringent decarbonization targets and/or more costly domestic abatement opportunities. In the absence of international emissions trading, the regions pursuing A/R are the same as when there is trading, but the overall level of A/R is lower. In contrast, without trading, more regions pursue BECCS compared to when trading is allowed, but the overall level of BECCS is substantially reduced. Countries with lower levels of CDR adoption have low biomass yields and/or high cropland prices, or limited land availability.

We find that with both CDR and international emissions trade (CDR_Trade case), the largest buyers of offsets, in the long run, are China, India, Europe, and the USA (Fig. 3). In the NoCDR_Trade case, however, China and India are important sellers of emission permits in the first half of the century, with China switching to a buyer in the second half of the century. The USA and Europe are still main buyers, and Africa a main seller in the NoCDR_Trade case. While the relative stringency of the assumed regional emissions caps is one factor determining which regions are buyers or sellers of permits, other factors, such as land availability and the regional cost of CDR and other mitigation options, are also important. As a test, under the CDR_Trade case we increased China's emissions cap, allowing 25% more emissions cumulatively over the century (see Supplementary Figs. 15 and 16). In this case, despite having a less stringent emissions cap than other regions, China remains a buyer of emissions for most of the century because offsets from A/R or BECCS are still cheaper than domestic abatement options.

### Economic impacts, transfers, and CDR value
Carbon prices depend on the availability of CDR and international emissions trading. The relationship between scenarios is more

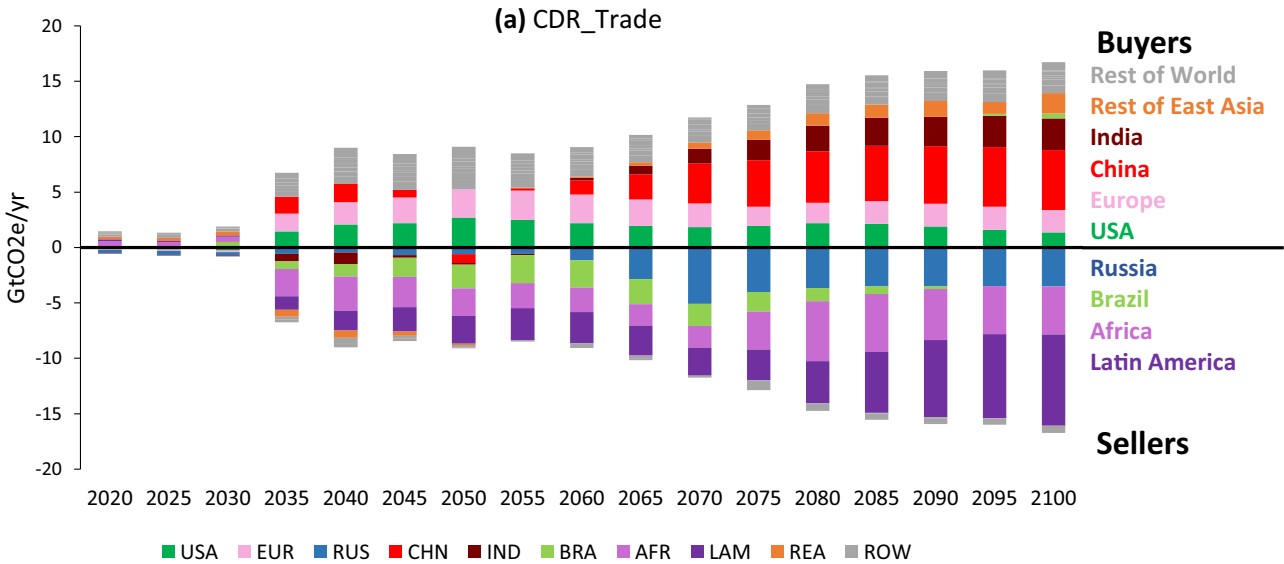

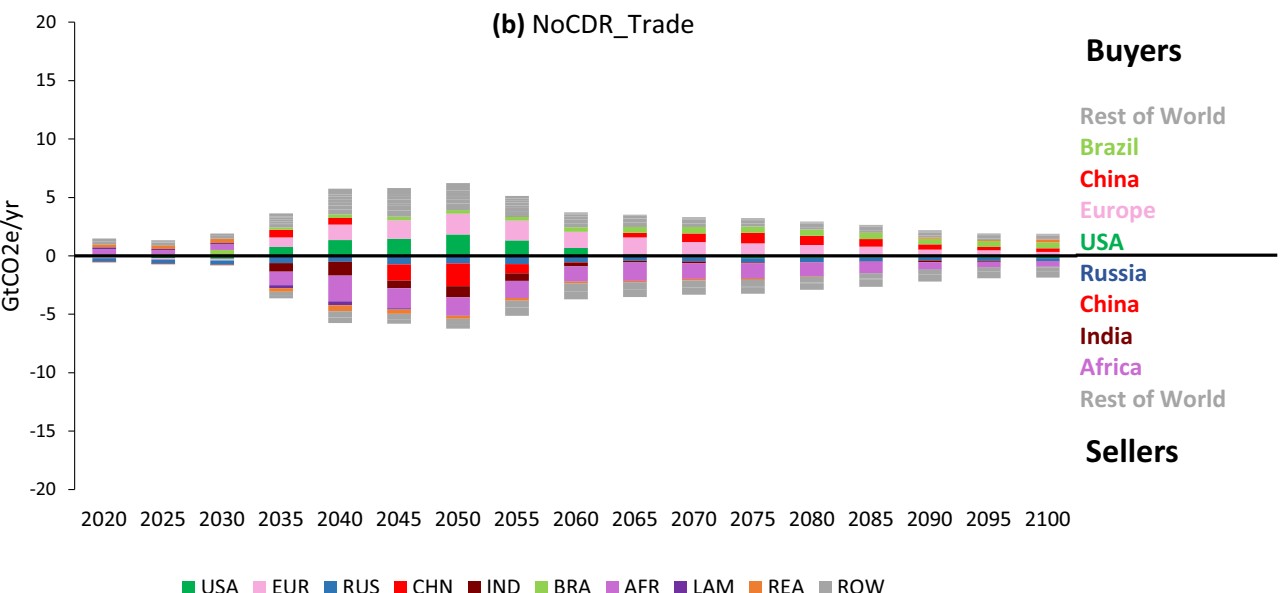

**Fig. 3 | Greenhouse gas (GHG) permits bought (positive values) and sold (negative values) by region in 2 °C scenarios allowing trade.** For a given region, permits bought is calculated as realized emissions minus the emissions cap. **a** CDR_Trade case and **b** NoCDR_Trade case. USA United States, EUR Europe (EU+), RUS Russia, CHN China, IND India, BRA Brazil, AFR Africa, LAM Other Latin America, REA Other East Asia, ROW Rest of World. Regional definitions are provided in Supplementary Fig. 1.

informative than absolute carbon prices, given uncertainty about mitigation options and costs. We find substantially higher carbon prices occurring when neither CDR nor international emissions trading is available (Fig. 4a). The lack of CDR and trade forces each region to meet its cap domestically without the use of negative emissions, which can be very expensive for some regions. With global trading and CDR, the optimal strategy for many regions is to emit above their cap and purchase offsets from the international market. This strategy results in the highest GDP growth for all countries (Fig. 4b). The availability of CDR without trading results in higher GDP than trading without CDR, and GDP growth is lowest when neither CDR nor trading is possible.

When CDR is allowed, the lower mitigation costs reflected in lower carbon prices increase GDP. Higher GDP, in turn, increases overall food consumption, and that higher demand increases crop prices. Trade in

GHG permits further increases GDP, consumption, and crop prices (Supplementary Fig. 6). In addition to this income effect, agricultural prices are also affected by land use—more land use changes in the scenarios with CDR (Supplementary Fig. 5) drive prices higher. However, despite higher agricultural prices, overall GDP is still improved with the deployment of CDR.

Across the scenarios, the scale of emissions traded and the scale of CDR both move in the opposite direction of the carbon prices. As a result, the total amount of resources spent on international offsets is similar with and without CDR, and reaches over $7 trillion (real 2007 USD) by 2100 (Fig. 5a). A growing portion of that spending comes from emerging economies. Although some countries have large expenses buying permits when trading is allowed, this strategy is less costly than not trading at all. Similarly, the total revenue received by CDR producers is similar with and without emissions trading and will reach over

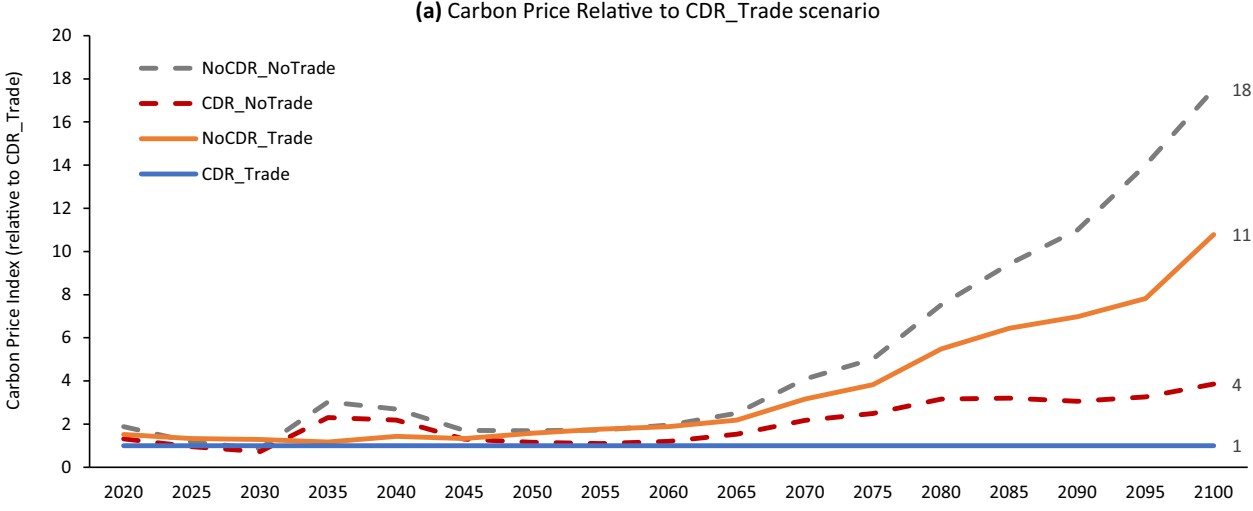

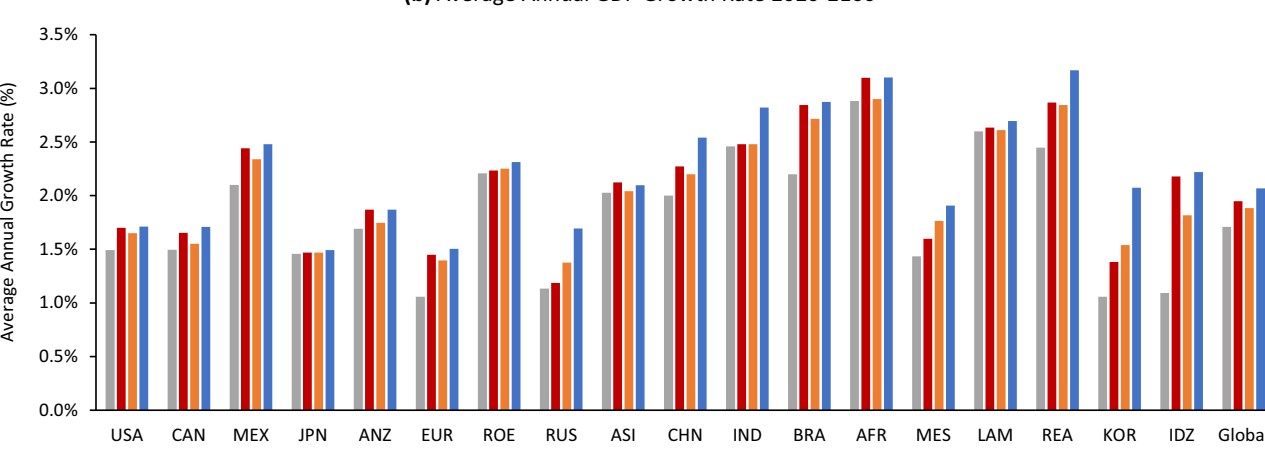

**Fig. 4 | Economic outcomes in the 2 °C scenarios. a** Carbon price index relative to the CDR_Trade scenario. For the scenarios without international trading, the carbon prices are reported as weighted averages of the regional carbon prices weighted by the regional cap at time $t$. The projected carbon price in the CDR_Trade scenario is about $130 in 2050 and $400 in 2100. **b** Regional average annual real GDP growth rates. USA United States, CAN Canada, MEX – Mexico, JPN Japan, ANZ Australia & New Zealand, EUR Europe (EU+), ROE Other Eurasia, RUS Russia, ASI Dynamic Asia (Malaysia, Philippines, Singapore, Taiwan), CHN China, IND India, BRA Brazil, AFR Africa, MES Middle East, LAM Other Latin America, REA Other East Asia, KOR Korea, IDZ Indonesia. Regional definitions are provided in Supplementary Fig. 1.

$9 trillion by 2100 (Fig. 5b). With trading, the vast majority of CDR revenue accrues in emerging economies. Importantly, while these scenarios involve similar amounts of resource transfers, the efficiency of the implied spending differs substantially, as evidenced by the differential implications for GDP. Spending is most efficient when both CDR and emissions trading are unconstrained. Under a more stringent climate mitigation scenario (-1.5 °C), the scale of offset spending and CDR revenue is higher, and the difference in CDR revenue under different trading assumptions is larger (Supplementary Figs. 12–14).

## Discussion

Results from our scenarios show the importance of policy design decisions, such as those related to Article 6 of the Paris Agreement, and technology availability, particularly CDR, in driving mitigation outcomes. The full economic potential of afforestation and BECCS is conditioned on international emissions trading as the comparative advantage in land-based CDR lies in a few regions. Regions with more stringent targets tend to use trade in permits to access less expensive negative emissions. Similarly, the potential for trade is conditioned on

CDR because, as net national emissions targets approach zero, trading can only occur when CDR enables some regions to retain net positive emissions. Thus, we find that CDR and international trade in emissions permits mutually reinforce each other. As such, international cooperation on mitigation efforts would enable greater use of CDR to attain climate stabilization targets at a lower cost.

The volume of permits traded and which regions are buying and selling depends not only on comparative advantages in A/R and BECCS but also on the assumed decarbonization trajectories for each country. In particular, we find some regions, such as China and India, switching from sellers to buyers as emissions constraints are tightened, which may create an incentive to resist strengthening targets in order to preserve the right to sell[45]. In addition, while the combination of international trade and CDR results in large resource transfers from permit buyers to sellers, most of those resources end up in a limited set of regions (e.g., Africa, Latin America, Brazil, and Russia), with emerging economies spending increasing amounts on offsets over time. Other distributional outcomes are possible, but such outcomes would require a more purposeful design of emissions caps or transfer

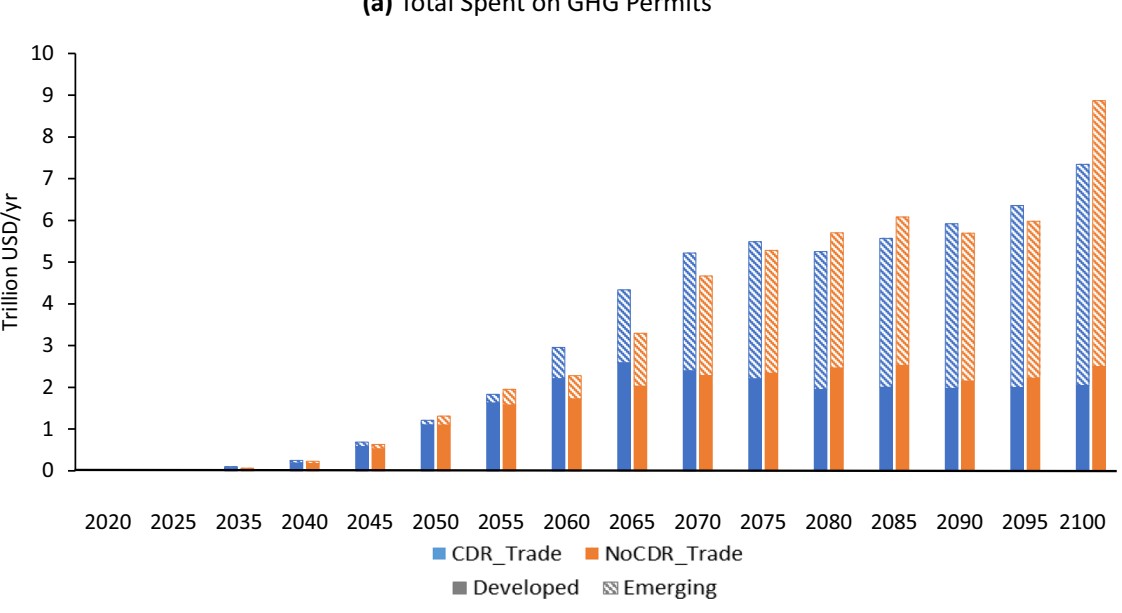

**(a)** Total Spent on GHG Permits

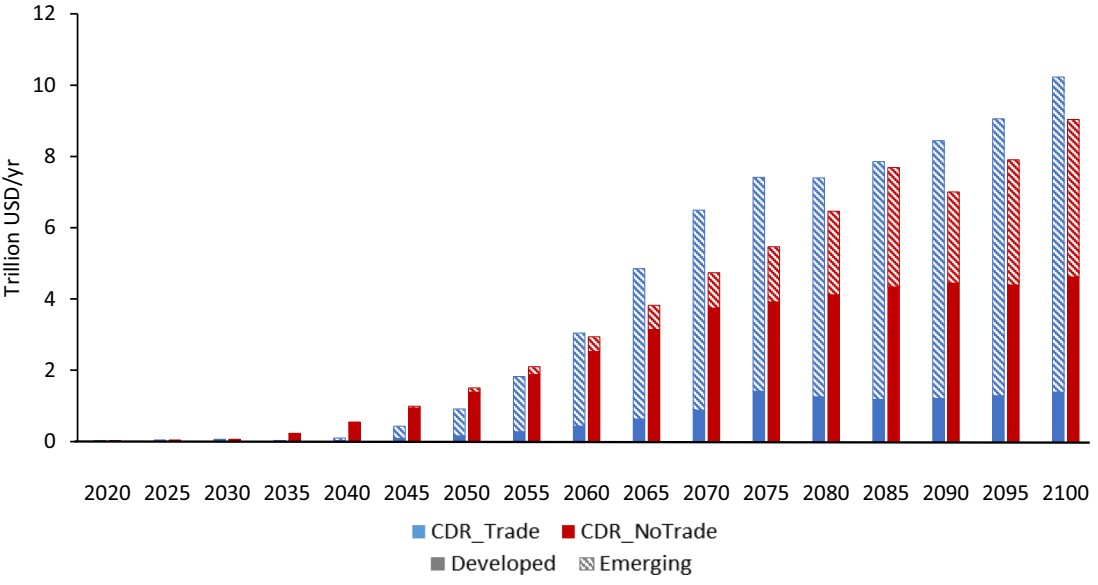

**(b)** Total Revenue Received by CDR Producers

**Fig. 5 | Transfers in the 2 °C scenarios. a** Total amount spent on greenhouse gas (GHG) permits, and **b** total revenue received by carbon dioxide removal (CDR) producers. Dollar values are real 2007 USD. See Supplementary Fig. 1 for definitions of developed and emerging regional groupings.

mechanisms. Similarly, the distribution between different economic and energy sectors could be managed in different ways, but that is not a focus of this study.

It is important to note that even when countries are buyers and spend more on offsets, our GDP results show that such a strategy is less costly than meeting the domestic target without trade. For most regions, though, and at the global scale, the availability of CDR has a larger positive impact on GDP than the availability of international emissions trading, as CDR enables substantially lower carbon prices. Still, employing CDR and international trading together maximizes GDP outcomes.

Our scenarios further suggest roles for both BECCS and A/R, with different options favored in different regions and at different points in time. Generally, A/R deploys before BECCS, with BECCS deploying

later in the century when carbon prices are higher. Countries most adopting BECCS and A/R have comparative advantages in producing biomass, higher land availability, and relatively lower cropland prices.

Future research might consider a broader set of land-based CDR options, such as biochar and carbon sequestration in agricultural soils, as well as other CDR options, such as DAC. Unless DAC costs decline substantially, BECCS would deploy at a lower carbon price than DAC[46], so we would not expect the inclusion of DAC to alter our results unless other options were limited. For example, in the absence of international emissions trading, some regions with unfavorable prospects for BECCS may adopt DAC instead. Future work could also investigate the impacts of alternative approaches for addressing differences in the permanence of different CDR options within emission trading systems. The main scenarios in this paper treat A/R and BECCS equally in terms

of CDR crediting. A risk associated with equal crediting is that allowing non-permanent CDR (such as from A/R) to offset energy system emissions could result in higher total emissions later if the CDR is reversed (e.g., if forests are cleared or burned in the future). However, we expect the overall finding that CDR and emissions trading reinforce one another to be robust to considerations of alternative CDR options and crediting approaches.

## Methods

### The Economic Projection and Policy Analysis (EPPA) model

We deploy a multi-region, multi-sector dynamic model of the global economy, the EPPA model, which represents the market interactions among all sectors and regions of the world[47,48]. The model projects the behavior of economic agents in optimizing production and consumption decisions, taking into account the resources available (labor, capital, land, natural resources), market prices, and alternative policies in place (such as those constraining greenhouse gas emissions). EPPA combines economic data and physical information on natural resources and greenhouse gas emissions. Economic data is provided by social accounting matrices depicting the structure of regional economies and includes bilateral trade and energy markets in physical units from the Global Trade Analysis Project database[49], organized into 18 regions and 14 sectors. Data on greenhouse gases (GHG) are obtained from several sources, including the International Energy Agency[50] for $CO_2$ emissions from energy consumption, Boden et al.[51] for emissions from cement production, and the Emissions Database for Global Atmospheric Research (EDGAR) Version 4.2[52] and Bond et al.[53] for non-$CO_2$ GHGs and conventional air pollutants.

A broad set of energy sources are available in EPPA, including biomass technologies providing liquid fuels and power generation. Biomass production competes for land with other uses, such as food and forestry production. Land use changes among alternative land use categories are endogenously represented. Natural vegetation (natural forest and natural grasslands) may be converted to agricultural uses, and agricultural land can be reverted to natural states (secondary vegetation).

The model projects future economic pathways in 5-year intervals from 2025 to 2100. Economic development through 2020 is benchmarked to historical data and short-term GDP projections of the International Monetary Fund (IMF)[54]. Future projections are driven by economic growth resulting from decisions related to savings and investment and exogenously specified productivity improvements in labor, capital, land, and energy. Demand for goods and services increases over time due to GDP and income growth. Higher cost grades of depletable resources are accessed as lower cost stocks are depleted. Backstop and advanced technologies may become cost competitive as conventional energy sources become more expensive. These economic drivers, combined with imposed policies and shocks, determine the economic trajectories over time and across scenarios.

### Land use changes in EPPA

Five broad land use categories are represented in EPPA: cropland, pasture, managed forest, natural forests and natural grassland. The base year land areas are built from the GTAP8 Land Use and Land Cover Database[55], from FAOSTAT production data and cropland and pasture data from Ramankutty[56], and the Terrestrial Ecosystem Model[57,58], which uses historical land use transitions from Hurtt et al.[59]. Land use conversion explicitly considers conversion costs, such as improvements (draining, tilling, fertilization, fencing) converting pastureland to cropland, or forestland to pastureland or cropland. A land supply response based on land conversions observed over the past few decades is used to calibrate the conversion of natural areas to agricultural use, representing factors that may prevent land conversion, such as increasing costs associated with larger deforestation in a single period and institutional costs (such as limits on deforestation, public

pressures for conservation, or establishment of conservation easements or land trusts). Land supply elasticities are based on values reviewed in the literature[60]. Initial rents on natural forests are calculated using a non-use value approach combining the cost of access to remote timber land and data from an optimal timber harvest model for each region and timber type[61].

Following estimates by Ray et al.[62], we assume exogenous technological changes in land yields, reflecting assessments of potential productivity improvements. In addition to exogenous yield changes and land conversion, agricultural output can grow by intensification of land use through partial substitution of other inputs and other primary factors in the agricultural production functions as relative prices change over time.

Assuming equilibrium in the base year, conversion costs from one land use category to another accounts for their differences in value. Natural forest and natural grass areas may be converted to other uses or conserved in their natural state. The reservation value of natural lands enters each regional representative agent welfare function with an elasticity of substitution with other consumption goods and services. Hence, the value the agent derives from natural land itself, is a deterrent to conversion. $CO_2$ emissions are accounted for when forest areas are converted to other uses with lower carbon density. Average carbon stocks above and below ground in each land use type are included for each region of EPPA using information from the Terrestrial Ecosystem Model[57,58].

The land use transformation approach in EPPA is an alternative from the common Constant Elasticity of Transformation (CET) approach often used in computable general equilibrium models. It allows for higher flexibility in land use changes expected in longer term analysis, explicitly accounts for conversion costs, and keeps consistency in accounting physical units of land, preserving total area—all features not observed in the CET approach[63].

Primary agricultural, livestock, and forestry products are mainly demanded as inputs for food, energy, and other sectors of the economy. Food and agriculture expenditure shares decrease as total household consumption increases due to income elasticities less than one, which is represented in EPPA by Stone-Geary preference systems[47].

Land use changes in EPPA capture several mechanisms of agriculture markets, including changes in the extensive margin (land conversion), changes in the intensive margin (land intensification through substitution of other inputs for land), demand responses due to price changes, shifts among crop, livestock, and forestry products, redistribution of crops among regional and global cropland area through international trade.

### BECCS and afforestation in EPPA

BECCS is explicitly represented in EPPA as a source of electricity and as a provider of carbon credits through capturing and storing $CO_2$ emissions from biomass combustion. BECCS is parameterized considering capital and labor costs, as well as costs of energy, land, materials and other inputs, and biomass transportation[64]. EPPA also captures bio-crop production, biomass conversion to electricity with $CO_2$ capture, transport and underground storage of $CO_2$, and the competition of BECCS with other low-carbon technologies. Economic drivers determine the deployment of BECCS in EPPA[10]. Political or sustainability constraints (e.g., water use or biodiversity loss) are not considered in the model. However, policies imposing limits on $CO_2$ emissions from land use changes may indirectly limit or prevent the conversion of natural forest areas to biomass production.

Afforestation and reforestation are allowed in land use transitions in EPPA, through the conversion of cropland and managed forests to natural (secondary) forests. The costs of conversion are explicitly captured by the difference in land value between the natural forest area and the land category being converted. In the base year, the value

of natural forest areas is usually lower than the value of agricultural areas, which means that the opportunity cost of the agricultural area prevents afforestation from happening. However, afforestation leads to carbon sequestration, which can become a valuable output if a carbon policy allowing such offsets is imposed. The amount of carbon provided by converting agricultural areas back to natural forests will depend on the regional carbon density of these areas. Carbon is sequestered through time following a linear approximation of the average growth period of natural forests in each region, and it stops when forests reach a maturation age. Forest owners receive the carbon price in the period in which afforestation begins, proportional to the carbon sequestered in the first five years of the project. The total potential forest area in each EPPA region is limited by ecological conditions, based on Griscom et al.[18] and Roe et al.[65].

## Scenarios

We developed four main scenarios, all of which attain the same 2 °C temperature stabilization target:

- NoCDR_NoTrade: BECCS and A/R are not available; there is no international trade in emissions permits
- NoCDR_Trade: BECCS and A/R are not available; there is international trade in emissions permits
- CDR_NoTrade: BECCS and A/R are available; there is no international trade in emissions permits
- CDR_Trade: BECCS and A/R are available; there is international trade in emissions permits

All scenarios follow a global GHG emissions pathway that was designed to limit the end-of-century increase in temperature to 2 °C with a 66% likelihood given an assessment of uncertainty in climate response to greenhouse gas forcing in the MIT Earth System Model[66]. Regional emissions caps were constructed to be consistent with the global emissions pathway, with differentiated responsibilities across regions based on existing commitments and proportional historical contributions to GHG concentrations. All countries have emissions targets consistent with their NDCs to 2030. Beyond that, we assume different years for peak emissions and different rates of decarbonization for G20 economies (except for India) vs. other developing economies vs. India. The assumed regional GHG emissions constraints, along with regional emissions trajectories from a reference no policy scenario, are shown in Supplementary Fig. 2. In the Supplementary Information, we consider a range of sensitivity cases.

Within a given region, the emissions target path effectively represents an economy-wide cap-and-trade system. In such a system, the allocation of abatement across sectors is determined endogenously such that marginal abatement costs across sectors are harmonized. Consequently, in this study, CDR deployment in one sector (for example, BECCS in power) may be used to offset emissions in another sector (for example, emissions from remaining fuels used in end uses or non-$CO_2$ emissions). The extent to which permit transfers actually occur across sectors to achieve these outcomes depends in part on how sectoral obligations are specified. The model outcomes discussed in this paper do not depend on these policy design details.

### Reporting summary

Further information on research design is available in the Nature Portfolio Reporting Summary linked to this article.

## Data availability

The data generated in this study are provided in the Source Data file. Source data are provided with this paper.

## Code availability

A publicly available version of the Economic Projection and Policy Analysis (EPPA) model is available at: https://globalchange.mit.edu/ research/research-tools/human-system-model/download. The version of the model applied here includes an updated and expanded representation of technology options (including bioenergy with carbon capture and storage and afforestation/reforestation), as well as updates to represent endogenous land use changes and associated emissions. The model is written in GAMS (General Algebraic Model System software, https://www.gams.com). The underlying base year model data is from the Global Trade Analysis Project (GTAP), Version 8 (https://www.gtap.agecon.purdue.edu/databases/default.asp).

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

## Acknowledgements

This work received funding support from the ExxonMobil Technology and Engineering Company. The views presented in this paper are those of the authors alone. The EPPA model employed in the analysis is supported by an international consortium of government, industry, and foundation sponsors of the MIT Joint Program on the Science and Policy of Global Change (see the list at: https://globalchange.mit.edu/sponsors/current).

## Author contributions

J.M. and A.G. advanced the model, performed the model simulations, and carried out the analysis. B.K.M., H.K. and S.P. provided critical input on the study design. J.M. took the lead in drafting the manuscript. J.M., A.G., B.K.M., H.K. and S.P. contributed to the interpretation and visualization of model results, provided critical feedback, and helped shape the final manuscript.

## Competing interests

The authors declare no competing interests.
