## [Peer Review File · Nature Communications]

Mutual Reinforcement of Land-based Carbon Dioxide Removal and International Emissions Trading in Deep Decarbonization ScenariosReviewer #1 (Remarks to the Author):

The articles focus of exploring carbon dioxide removals (CDR) in long term climate scenarios with and without trade between regions. With Article 6 of the Paris agreement currently being negotiated and an increased international focus on net-zero targets, this is a highly relevant topic. This is also an area where there have been limited research.

The most noteworthy result from the paper is the exploration of the connection between CDR deployment and trade. The results show that allowing for trade between regions significantly increases the amount of CDR, while allowing for higher GDP growth. If one has some background knowledge about biomass potentials it is not surprising that allowing for trade between regions increases land-based CDR deployment, but the article explores the connection, different consequences, (including GDP growth), and details (such as CDR supply and demand in different regions) of allowing, or not allowing, trade, in a structured way.

The article is well formulated and present the results and arguments in a structured and clear way, with informative figures. The methods section and supporting material provide good descriptions of the methods.

Major comment:

Even though the article over all is of high quality I have one major concern, and this is regarding the included CDR methods. The model includes two types of CDR: bioenergy with carbon capture and storage (BECCS) and afforestation and reforestation (A/R). The two types have different permanence, where BECCS provides "permanent" storage while A/R may be reversed through disturbances (such as fires, disease or exploitation). These differences are acknowledged in the introduction, but then CDR permits from the two types are treated equally in the model (unless I have misunderstood something).

There is a growing discussion about how CDR methods with varying levels of permanence should be handled, including the proposed "like-for-like" principle, which suggest that the durability of the removal should be matched to the permanence of the emission. (One article where this is discussed is: "Secure robust carbon dioxide removal policy through credible certification", by Schenuit et al, in Communications Earth & Environment, 2023).

There is a risk with scenarios where credits from non-permanent CDR (such as A/R) are allowed to balance fossil fuel emissions, as this allows more total fossil fuel emissions while the emissions stored in forests could later be reversed (the risk for this could even increase with climate change, as heat waves and extreme weather will become more common). This is for example seen in the Fig.S3 in the supplementary material where the CO₂ Fossil emissions are much higher in the CDR_Trade scenario compared to the NoCDR_Trade scenario, around 2050-2070 when CDR from A/R (named CO₂_LUC in the figure) are high.

The current debate about how to handle CDR methods with different permanence should be mentioned in the introduction. Optimally the issue should be addressed in the model, by for example including a "like-for-like" scenario, where credits from A/R are only allowed to counterbalance appropriate emissions (such as short-lived GHG and/or LUC emissions). This would increase the relevance of the article, as it would also contribute to the discussion on how to handle permanence for different CDR methods.

If this is not possible, the issue of differing permanence between A/R and BECCS should be discussed in relation to the results and the risks and limitations with the approach should be made clear.

Minor comments:

- Unless it is against the policy of the journal I would suggest to briefly mention that your study is based on modelling already in the abstract (would only need to be a few extra words). To make it

easier for readers to know what to expect from the paper and what the conclusions are based on.

- You write that "However, there is no discussion in the literature about the interactions between international permit trade and land-based CDR." I did however find one article on the topic: "Fajardy, Mathilde, and Niall Mac Dowell. "Recognizing the value of collaboration in delivering carbon dioxide removal.", which focuses on BECCS and trade between regions. So you may want to add that reference and comment on it (either in the introduction or discussion).
- It would strengthen the results to compare the levels of CDR that are deployed in your model to the ones deployed in other studies (for example potential studies for CDR, or other modelling studies).
- In the supplementary material Figure S5 show global changes in land use. As the article focuses on land-based CDR land-use is an important issue, but this figure is never referenced in the main article.

Reviewer #2 (Remarks to the Author):

The paper uses a well-established Computable General Equilibrium Model (CGE) that includes land-use changes and afforestation and BECCs as negative emission technologies (NETSs) to analyze different scenarios in which the 2°C target is met. The scenarios differ with respect to whether there is international emission trading to reach the national / regional targets derived from the NDCs and whether or not also removal units can be traded. This results in a 2x2 matrix of four scenarios. Generally, the scenarios are straight forward and something one would naturally analyze with a CGE mode. The resulting efficiency gains are what one would expect and not overly surprising, but the results on timing and location of CDR measures and abatement are of interest. Furthermore, such analysis has obviously not been undertaken so-far and given the importance of both NETs and keeping the costs of meeting the Paris temperature targets low, it is certainly relevant for a broader audience, such as is addressed in Nature Communications. Also, the paper is well written and concise and I have no doubts that it is based on sound analysis. I have only some small comments that should be addressed before the paper can be published.

Article 6 of UNFCCC is only very briefly mentioned. Please elaborate a little more on how NETs / NETs trading could be included in the international framework. This will also stress the relevance of your work.

I miss some relevant literature in the brief literature review:

On the relevance of trading for cost efficiency to reach the NDCs: Böhringer et al. (2021). Climate policies after Paris: Pledge, trade & recycle.

Overview on trading of NET credits: Michaelowa et al. (2023). International Carbon Markets for carbon dioxide removal.

Starting from [Rickels et al. (2012). Economic prospects of ocean iron fertilization in an international carbon market] there should also be a brief discussion of existing literature of model-based analysis of NET trading.

Briefly mention in the main text how the global net emission path until 2100 looks like, especially it is relevant when net zero will be reached,

If these additions require space, one could move Fig 5 to the appendix and just mention some of the numbers in the text

Reviewer #3 (Remarks to the Author):

Please find attached my review comments.

Report for “Mutual Reinforcement of Land-based Carbon Dioxide Removal and International Emissions Trading in Deep Decarbonization Scenarios”, Manuscript Number NCOMMS-23-53045-T

The paper investigates the interaction between land-based Carbon Dioxide Removal (CDR) options and international trade in GHG permits, using the Economic Projection and Policy Analysis (EPPA) model. This model, a multi-region, multi-sector recursive dynamic model of the global economy, comprehensively captures various aspects. These include the costs of bioenergy with carbon capture and storage (BECCS) and biomass technologies, land conversion costs, CO₂ emissions and sequestration from direct and indirect land use changes, land availability and competition, international trade of agricultural and food commodities, and decarbonization policies, including their effects on technology competition, goods prices, and aggregate income and consumption.

Specifically, they evaluate afforestation and reforestation (A/R) and BECCS as CDR options to assess the potential deployment of such technologies and the size of international carbon markets. The findings indicate that CDR and international emissions trading complement each other in deep decarbonization scenarios (i.e., 2°C temperature stabilization target). This is because CDR potential is not evenly distributed geographically (allowing trade to unlock this potential) and because trading in a net-zero emissions world requires negative emissions (allowing CDR to enable trade).

For example, because of permit trade, Latin America, Africa and Russia implements significantly more BECCS as the US, Europe, China and India buy GHG permits, which happens in 2060 onward. In contrast, without permit trade, a greater variety of countries pursue BECCS, albeit at a much smaller scale, as permits cannot be sold. Afforestation and reforestation are primarily implemented by the same countries implementing BECCS, with Brazil joining the group. However, the peak implementation occurs around 2060, and with permit trade, the implementation level is substantially higher, albeit with no change in the adopting countries.

I really enjoyed reading this paper. It is well-written and covers a relevant topic. The findings are interesting, and the contribution to the literature is substantial. Thus, I have only a few minor suggestions for improvement.

Main points (not in order of their significance):

1. In the abstract, the following part is unclear without reading about Fig. 5: “because carbon prices are lower when emissions trading and CDR deployment are higher, the total amount spent on offsets and the revenue received by CDR producers does not vary strongly with the amount of emissions trading or CDR.” As I believe the word limit of the abstract is 200 words, the authors could provide a clearer description with additional words.
2. I understand that this paper focuses on the interaction between international permit trade and land-based CDR and that there are many studies examining the role of international trade. Is there a similar study focusing on CDR (e.g., with or without international permit trade)? If yes, we would appreciate knowing the reference(s). If not, even if we think of only the case with international permit trade, the paper still makes a significant contribution, and it can be pointed out.

Minor comments:

1. Biochar and direct air capture (DAC) are deferred for the future study. Although mentioned in the final paragraph of the Discussion section, it would be better to point out already in the introduction that BECCS would likely be deployed at a lower cost compared to DAC.
2. The small table within Fig. 1 would be better with a consistent format with Table 1, unless there is a specific reason for the switch in formatting between row vs. column.
3. In Figure 4a, the decision to display the index relative to carbon price in the CDR_Trade scenario rather than the carbon price itself may enhance graph comprehension. However, some readers may desire to know the carbon price directly for each scenario. Including such information in the supplementary material could be beneficial.

Response to Reviewer Comments

We thank the reviewers for their thoughtful comments. We provide a point-by-point response to the all comments below in red. We believe the resulting revisions have improved the paper.

REVIEWER COMMENTS

Reviewer #1 (Remarks to the Author):

The articles focus of exploring carbon dioxide removals (CDR) in long term climate scenarios with and without trade between regions. With Article 6 of the Paris agreement currently being negotiated and an increased international focus on net-zero targets, this is a highly relevant topic. This is also an area where there have been limited research.

The most noteworthy result from the paper is the exploration of the connection between CDR deployment and trade. The results show that allowing for trade between regions significantly increases the amount of CDR, while allowing for higher GDP growth. If one has some background knowledge about biomass potentials it is not surprising that allowing for trade between regions increases land-based CDR deployment, but the article explores the connection, different consequences, (including GDP growth), and details (such as CDR supply and demand in different regions) of allowing, or not allowing, trade, in a structured way.

The article is well formulated and present the results and arguments in a structured and clear way, with informative figures. The methods section and supporting material provide good descriptions of the methods.

Thank you for the helpful review. We have responded to specific comments and suggestions below.

Major comment:

Even though the article over all is of high quality I have one major concern, and this is regarding the included CDR methods. The model includes two types of CDR: bioenergy with carbon capture and storage (BECCS) and afforestation and reforestation (A/R). The two types have different permanence, where BECCS provides “permanent” storage while A/R may be reversed through disturbances (such as fires, disease or exploitation). These differences are acknowledged in the introduction, but then CDR permits from the two types are treated equally in the model (unless I have misunderstood something).

There is a growing discussion about how CDR methods with varying levels of permanence should be handled, including the proposed “like-for-like” principle, which suggest that the durability of the removal should be matched to the permanence of the emission. (One article where this is discussed is: “Secure robust carbon dioxide removal policy through credible certification”, by Schenuit et al, in Communications Earth & Environment, 2023).

There is a risk with scenarios where credits from non-permanent CDR (such as A/R) are allowed to balance fossil fuel emissions, as this allows more total fossil fuel emissions while the emissions stored in

forests could later be reversed (the risk for this could even increase with climate change, as heat waves and extreme weather will become more common). This is for example seen in the Fig.S3 in the supplementary material where the CO₂_Fossil emissions are much higher in the CDR_Trade scenario compared to the NoCDR_Trade scenario, around 2050-2070 when CDR from A/R (named CO₂_LUC in the figure) are high.

The current debate about how to handle CDR methods with different permanence should be mentioned in the introduction. Optimally the issue should be addressed in the model, by for example including a “like-for-like” scenario, where credits from A/R are only allowed to counterbalance appropriate emissions (such as short-lived GHG and/or LUC emissions). This would increase the relevance of the article, as it would also contribute to the discussion on how to handle permanence for different CDR methods.

If this is not possible, the issue of differing permanence between A/R and BECCS should be discussed in relation to the results and the risks and limitations with the approach should be made clear.

We agree the debate about how to handle differences in the permanence of CDR options is an important one. We have now mentioned it in the introduction and raise it again in the results section. While our paper is not focused on exploring different crediting options (e.g. “like-for-like”, discounting, etc.), we have included scenarios that provide some insight into the bounds of different crediting options: the main “CDR” scenarios assume equal permits for A/R and BECCS and an additional set of “BECCSonly” scenarios assume no credits for A/R. We included additional figures comparing the “CDR” and “BECCSonly” scenarios in the Supplementary Information and reference them in a new paragraph in the results section. We show that the assumption about how to handle A/R offsets does not change the relationship that is the focus of the paper—that between land-based CDR deployment in general and the level of emissions trading.

Minor comments:

- Unless it is against the policy of the journal I would suggest to briefly mention that your study is based on modelling already in the abstract (would only need to be a few extra words). To make it easier for readers to know what to expect from the paper and what the conclusions are based on.

Done.

- You write that “However, there is no discussion in the literature about the interactions between international permit trade and land-based CDR.” I did however find one article on the topic: "Fajardy, Mathilde, and Niall Mac Dowell. "Recognizing the value of collaboration in delivering carbon dioxide removal.", which focuses on BECCS and trade between regions. So you may want to add that reference and comment on it (either in the introduction or discussion).

Thank you for calling attention to this paper. We have cited it in the introduction (adjusting the referenced sentence accordingly) and also raise it in the results section.

- It would strengthen the results to compare the levels of CDR that are deployed in your model to the ones deployed in other studies (for example potential studies for CDR, or other modelling studies).

We have added comparisons to CDR levels in other studies in the Results section.

- In the supplementary material Figure S5 show global changes in land use. As the article focuses on land-based CDR land-use is an important issue, but this figure is never referenced in the main article. **Reference to figure S5 is now made in the “Economic impacts, transfers and CDR value” subsection of the Results section.**

Reviewer #2 (Remarks to the Author):

The paper uses a well-established Computable General Equilibrium Model (CGE) that includes land-use changes and afforestation and BECCs as negative emission technologies (NETs) to analyze different scenarios in which the 2°C target is met. The scenarios differ with respect to whether there is international emission trading to reach the national / regional targets derived from the NDCs and whether or not also removal units can be traded. This results in a 2x2 matrix of four scenarios. Generally, the scenarios are straight forward and something one would naturally analyze with a CGE mode. The resulting efficiency gains are what one would expect and not overly surprising, but the results on timing and location of CDR measures and abatement are of interest. Furthermore, such analysis has obviously not been undertaken so-far and given the importance of both NETs and keeping the costs of meeting the Paris temperature targets low, it is certainly relevant for a broader audience, such as is addressed in Nature Communications. Also, the paper is well written and concise and I have no doubts that it is based on sound analysis. I have only some small comments that should be addressed before the paper can be published.

Thank you for the thoughtful review. We have responded to the specific suggestions and comments below.

Article 6 of UNFCCC is only very briefly mentioned. Please elaborate a little more on how NETs / NETs trading could be included in the international framework. This will also stress the relevance of your work.

We have elaborated on this in the introduction.

I miss some relevant literature in the brief literature review:

On the relevance of trading for cost efficiency to reach the NDCs: Böhringer et al. (2021). Climate policies after Paris: Pledge, trade & recycle.

We have added reference to this paper.

Overview on trading of NET credits: Michaelowa et al. (2023). International Carbon Markets for carbon dioxide removal.

Thank you for drawing attention to this paper, we have added reference to it in the introduction.

Starting from [Rickels et al. (2012). Economic prospects of ocean iron fertilization in an international

carbon market] there should also be a brief discussion of existing literature of model-based analysis of NET trading.

We have added a short paragraph in the introduction to reference studies of different CDR / NET options.

Briefly mention in the main text how the global net emission path until 2100 looks like, especially it is relevant when net zero will be reached,

In the main text we now call attention to Supplementary Fig. 3 which shows the global net GHG emissions path.

If these additions require space, one could move Fig 5 to the appendix and just mention some of the numbers in the text

Reviewer #3 (Remarks to the Author):

The paper investigates the interaction between land-based Carbon Dioxide Removal (CDR) options and international trade in GHG permits, using the Economic Projection and Policy Analysis (EPPA) model. This model, a multi-region, multi-sector recursive dynamic model of the global economy, comprehensively captures various aspects. These include the costs of bioenergy with carbon capture and storage (BECCS) and biomass technologies, land conversion costs, CO₂ emissions and sequestration from direct and indirect land use changes, land availability and competition, international trade of agricultural and food commodities, and decarbonization policies, including their effects on technology competition, goods prices, and aggregate income and consumption.

Specifically, they evaluate afforestation and reforestation (A/R) and BECCS as CDR options to assess the potential deployment of such technologies and the size of international carbon markets. The findings indicate that CDR and international emissions trading complement each other in deep decarbonization scenarios (i.e., 2°C temperature stabilization target). This is because CDR potential is not evenly distributed geographically (allowing trade to unlock this potential) and because trading in a net-zero emissions world requires negative emissions (allowing CDR to enable trade).

For example, because of permit trade, Latin America, Africa and Russia implements significantly more BECCS as the US, Europe, China and India buy GHG permits, which happens in 2060 onward. In contrast, without permit trade, a greater variety of countries pursue BECCS, albeit at a much smaller scale, as permits cannot be sold. Afforestation and reforestation are primarily implemented by the same countries implementing BECCS, with Brazil joining the group. However, the peak implementation occurs around 2060, and with permit trade, the implementation level is substantially higher, albeit with no change in the adopting countries.

I really enjoyed reading this paper. It is well-written and covers a relevant topic. The findings are interesting, and the contribution to the literature is substantial. Thus, I have only a few minor suggestions for improvement.

Thank you for the review of this manuscript and for noting its scientific value.

Main points (not in order of their significance):

1. In the abstract, the following part is unclear without reading about Fig. 5: “because carbon prices are lower when emissions trading and CDR deployment are higher, the total amount spent on offsets and the revenue received by CDR producers does not vary strongly with the amount of emissions trading or CDR.” As I believe the word limit of the abstract is 200 words, the authors could provide a clearer description with additional words.
We tried to make this clearer in the abstract, but we are limited to 150 words.
2. I understand that this paper focuses on the interaction between international permit trade and land-based CDR and that there are many studies examining the role of international trade. Is there a similar study focusing on CDR (e.g., with or without international permit trade)? If yes, we would appreciate knowing the reference(s). If not, even if we think of only the case with international permit trade, the paper still makes a significant contribution, and it can be pointed out.
We have added a short paragraph in the introduction to reference CDR studies and also added reference to Fajardy and Mac Dowell (2020), which explicitly looked at the impact of inter-regional trading on BECCS.

Minor comments:

1. Biochar and direct air capture (DAC) are deferred for the future study. Although mentioned in the final paragraph of the Discussion section, it would be better to point out already in the introduction that BECCS would likely be deployed at a lower cost compared to DAC.
We have made note of this in the introduction.
2. The small table within Fig. 1 would be better with a consistent format with Table 1, unless there is a specific reason for the switch in formatting between row vs. column.
We switched the rows and columns to be consistent with Fig. 1
3. In Figure 4a, the decision to display the index relative to carbon price in the CDR_Trade scenario rather than the carbon price itself may enhance graph comprehension. However, some readers may desire to know the carbon price directly for each scenario. Including such information in the supplementary material could be beneficial.
In the caption of Figure 4 we added the projected carbon prices for 2050 and 2100 in the CDR_Trade scenario as points of reference (which can be used to estimate the absolute carbon prices in the other scenarios). In the related text we also note that the relationship between scenarios is more important than the absolute carbon prices since the specific numbers are driven by model assumptions about available mitigation options and costs.

Reviewer #1 (Remarks to the Author):

All of the minor comments have been addressed appropriately by the authors and I think that the revisions have improved the article.

I appreciate the new paragraph in the result section with references to figures comparing the "CDR" and "BECCSonly" scenarios in the Supplementary Information and that the issue of differing permanence between A/R and BECCS is raised clearer in the introduction.

In the Discussion there is a mention about the issue of permanence in the section about future research, which is good, but I would like to see 2-3 additional sentences, acknowledging that the main scenarios of the model treat A/R and BECCS "equally" when it comes to CDR permits, but that there is an issue of differing permanence between the two. Including some mentioning/discussion about the risk of allowing credits from non-permanent CDR (such as A/R) to balance fossil fuel emissions, as this allows more total fossil fuel emissions (while the emissions stored in forests could later be reversed). If this is added to the discussion, I deem the article fit for publication.

Response to Reviewer Comments

REVIEWER COMMENTS

All of the minor comments have been addressed appropriately by the authors and I think that the revisions have improved the article.

I appreciate the new paragraph in the result section with references to figures comparing the “CDR” and “BECCSonly” scenarios in the Supplementary Information and that the issue of differing permanence between A/R and BECCS is raised clearer in the introduction.

In the Discussion there is a mention about the issue of permanence in the section about future research, which is good, but I would like to see 2-3 additional sentences, acknowledging that the main scenarios of the model treat A/R and BECCS “equally” when it comes to CDR permits, but that there is an issue of differing permanence between the two. Including some mentioning/discussion about the risk of allowing credits from non-permanent CDR (such as A/R) to balance fossil fuel emissions, as this allows more total fossil fuel emissions (while the emissions stored in forests could later be reversed).

If this is added to the discussion, I deem the article fit for publication.

We thank the reviewer for their original comments and follow-up, and agree the revisions have improved the paper. We have followed the reviewer’s new suggestion to expand on the issue of permanence in the Discussion, adding the following two sentences:

“The main scenarios in this paper treat A/R and BECCS equally in terms of CDR crediting. A risk associated with equal crediting is that allowing non-permanent CDR (such as from A/R) to offset energy system emissions could result in higher total emissions later if the CDR is reversed (e.g. if forests are cleared or burned in the future).”